# Model-based deep embedding for constrained clustering analysis of single cell RNA-seq data

Tian Tian[1,4], Jie Zhang[2,4], Xiang Lin[2], Zhi Wei [2✉] & Hakon Hakonarson [1,3✉]

Clustering is a critical step in single cell-based studies. Most existing methods support unsupervised clustering without the a priori exploitation of any domain knowledge. When confronted by the high dimensionality and pervasive dropout events of scRNA-Seq data, purely unsupervised clustering methods may not produce biologically interpretable clusters, which complicates cell type assignment. In such cases, the only recourse is for the user to manually and repeatedly tweak clustering parameters until acceptable clusters are found. Consequently, the path to obtaining biologically meaningful clusters can be ad hoc and laborious. Here we report a principled clustering method named scDCC, that integrates domain knowledge into the clustering step. Experiments on various scRNA-seq datasets from thousands to tens of thousands of cells show that scDCC can significantly improve clustering performance, facilitating the interpretability of clusters and downstream analyses, such as cell type assignment.

[1] Center for Applied Genomics, The Children's Hospital of Philadelphia, Philadelphia, PA, USA. [2] Department of Computer Science, New Jersey Institute of Technology, Newark, NJ, USA. [3] Division of Human Genetics, Department of Pediatrics, The Perelman School of Medicine, University of Pennsylvania, Philadelphia, PA, USA. [4] These authors contributed equally: Tian Tian, Jie Zhang. ✉email: zhiwei@njit.edu; hakonarson@email.chop.edu

Clustering presents an essential data analysis and visualization tool that has become a key step in defining cell types based on the transcriptome and has emerged as one of the most powerful applications of scRNA-seq[1–3]. Early research applied traditional dimension reduction methods, such as PCA, t-SNE[4,5], UMAP[6], followed by k-means or hierarchical clustering to group and visualize cells, including SC3[7] (Spectral clustering), pcaReduce[8] (PCA + k-means + hierarchical), TSCAN[9] (PCA + Gaussian mixture model) and mpath[10] (Hierarchical), to name a few. However, unlike bulk RNA-seq or microarray, due to the extreme sparsity caused by dropouts and high variability in gene expression levels, traditional clustering approaches tend to deliver suboptimal results on scRNA-seq data sets[3,11].

Recently, various clustering methods have been proposed to overcome the challenges in scRNA-seq data analysis. Shared nearest neighbor (SNN)-Clip combines a quasi-clique-based clustering algorithm with the SNN-based similarity measure to automatically identify clusters in the high-dimensional and high-variable scRNA-seq data[12]. DendroSplit[13] applies "split" and "merge" operations on the dendrogram obtained from hierarchical clustering, which iteratively groups cells based on their pairwise distances (calculated upon selected genes), to uncover multiple levels of biologically meaningful populations with interpretable hyperparameters. If the dropout probability $P(u)$ is a decreasing function of the gene expression $u$, CIDR[14] uses a nonlinear least-squares regression to empirically estimate $P(u)$ and imputes the gene expressions with a weighted average to alleviate the impact of dropouts. Clustering analysis is performed on the first few principal coordinates, obtained through principal coordinate analysis (PCoA) on the imputed expression matrix[14]. SIMLR[15] and MPSSC[16] are both multiple kernel-based spectral clustering methods. Considering the complexities of the scRNA-seq data, multiple kernel functions can help to learn robust similarity measures that correspond to different informative representations of the data[17]. However, spectral clustering relies on the full graph Laplacian matrix, which is prohibitively expensive to compute and store[18]. The high complexity and limited scalability generally impede applying these methods to large scRNA-seq datasets[3].

The large number of cells profiled via scRNA-seq provides researchers with a unique opportunity to apply deep learning approaches to model the noisy and complex scRNA-seq data. scScope[19] and DCA[20] (Deep Count Autoencoder) apply regular autoencoders to denoise single-cell gene expression data and impute the missing values[21]. In autoencoders, the low-dimensional bottleneck layer enforces the encoder to learn only the essential latent representations and the decoding procedure ignores non-essential sources of variations of the expression data[21]. Compared to scScope, DCA explicitly models the over-dispersion and zero-inflation with a zero-inflated negative binomial (ZINB) model-based loss function and learns gene-specific parameters (mean, dispersion and dropout probability) from the scRNA-seq data. SCVI[22] and SCVIS[23] are variational autoencoders (VAE)[24] focusing on dimension reduction of scRNA-seq data. Unlike autoencoder, variational autoencoder assumes that latent representations learnt by the encoder follow a predefined distribution (typically a Gaussian distribution). SCVIS uses the Student's t-distributions to replace the regular MSE-loss (mean square error) VAE, while SCVI applies the ZINB-loss VAE to characterize scRNA-seq data. Variational autoencoder is a deep generative model, but the assumption of latent representations following a Gaussian distribution might introduce the over-regularization problem[25] and compromise its performance. More recently, Tian et al. developed a ZINB model-based deep clustering method (scDeepCluster)[26] and showed that it could

effectively characterize and cluster the discrete, over-dispersed and zero-inflated scRNA-seq count data. scDeepCluster combines the ZINB model-based autoencoder with the deep embedding clustering[27,28], which optimizes the latent feature learning and clustering simultaneously to achieve better clustering results.

Much of the downstream biological investigation relies on initial clustering results. Although clustering aims to explore and uncover new information (e.g., novel cell types), biologists expect to see some meaningful clusters that are consistent with their prior knowledge. In other words, totally exotic clustering with poor biological interpretability is puzzling, which is generally not desired by biologists. For a clustering algorithm, it is good to accommodate biological interpretability while minimizing clustering loss from computational aspect[3]. Most, if not all, existing algorithms for scRNA-seq, however, only support clustering in an unsupervised fashion, and are incapable of integrating prior information. If a method initially fails to find a meaningful solution, the only recourse may be for the user to manually and repeatedly tweak clustering parameters until sufficiently good clusters are found[3,29].

We note that prior knowledge has become widely available in many cases[30]. Quite a few cell type-specific signature sets have been published, such as Immunome[31], eTME[32]. Multi-omics sequencing data can also be used as prior knowledge, such as CITE-seq[33] (profiling single-cell transcriptome and surface proteins simultaneously) and single-cell ATAC-seq[34]. Alternatively, researchers could also define the markers based on pilot or cross-validation experiments[30]. Ignoring prior information may lead to suboptimal, unexpected, and even illogical clustering results. CellAssign[35] and SCINA[30] are two applications that have been proposed recently to leverage prior knowledge of cell-type marker genes. Their goal is to assign cells into one of the several predefined cell types. Each predefined cell type is described by some marker genes. CellAssign is essentially a generalized linear model in which latent cell types and marker genes together with other covariates to predict gene expression level. A marker gene is assumed to have an over-expression effect (to be estimated) relative to cells for which it is not a marker. CellAssign assumes that gene expression in terms of counts follows a negative binomial distribution (NB). SCINA uses a similar approach to utilize prior knowledge of marker genes, but differs from CellAssign by assuming that the normalized gene expressions follow a Gaussian distribution. Both CellAssign and SCINA have demonstrated significant improvement over existing methods, which supports the usage of prior knowledge.

However, there are several limitations of these methods. First, they are developed in the context of the marker genes and lack the flexibility to integrate other kinds of prior information. Second, they are only applicable to scenarios where cell types are predefined and well-studied marker genes exist. Poorly understood cell types would be invisible to these methods. Finally, they both ignore pervasive dropout events, a well-known problem for scRNA-seq data.

In this article, we are interested in integrating prior information into the modeling process to guide our deep learning model to simultaneously learn meaningful and desired latent representations and clusters. Unlike traditional hard-constrained clustering algorithms (e.g., COP K-means[36]), we convert (partial) prior knowledge into soft pairwise constraints and add them as additional terms into the loss function for optimization. The proposed method presents a more flexible form of semi-supervised clustering[37,38], and is more feasible in real scRNA-seq experiments. Here, we name the proposed model-based deep embedding clustering method as scDCC (Single Cell Deep Constrained Clustering). The network architecture of scDCC is summarized in Fig. 1. Basically, scDCC encodes prior knowledge

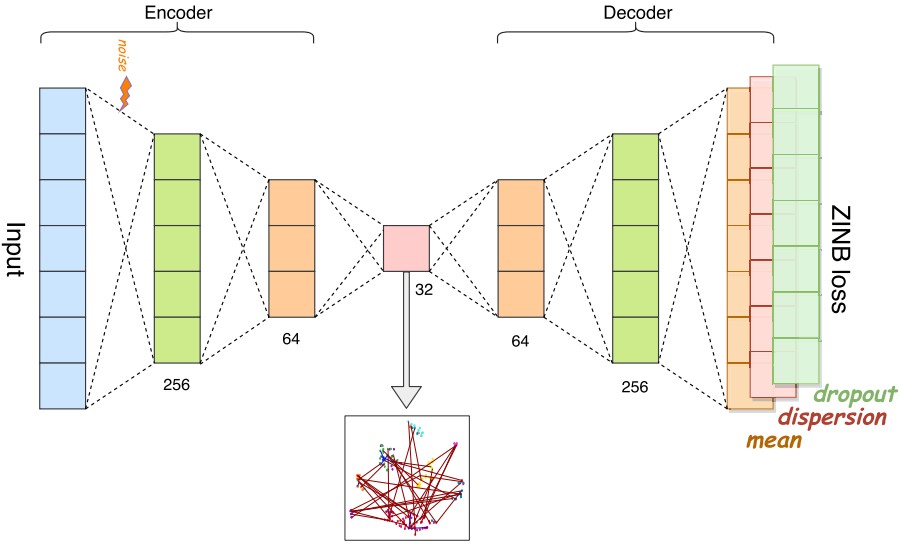

**Fig. 1 Network architecture of scDCC.** The autoencoder is a fully connected neural network. The number below each layer denotes the size of that layer.

**Table 1 Summary of four small scRNA-seq datasets.**

| Dataset | Sequencing platform | Sample size / #Cell | #Genes | #Groups |
|---|---|---|---|---|
| 10X PBMC | 10X | 4271 | 16,449 | 8 |
| Mouse bladder cells | Microwell-seq | 2746 | 19,079 | 16 |
| Worm neuron cells | sci-RNA-seq | 4186 | 11,955 | 10 |
| Human kidney cells | 10X | 5685 | 25,215 | 11 |

into constraint information, which is integrated to the clustering procedure via a novel loss function. We apply scDCC with pairwise constraints to the scRNA-seq datasets of various sizes (from thousands to tens of thousands of cells). Our extensive experimental results illustrate that domain knowledge can help to achieve better clustering performance under different scenarios. We expect that the downstream analysis, such as cell type assignment, will benefit from the prior knowledge and biologically meaningful clusters.

## Results

**Pairwise constraints.** Pairwise constraints mainly focus on the together or apart guidance as defined by prior information and domain knowledge. They enforce small divergence between pre-defined "similar" samples, while enlarging the difference between "dissimilar" instances. Researchers usually encode the together and apart information into must-link (ML) and cannot-link (CL) constraints, respectively[38,39]. With the proper setup, pairwise constraints have been proved to be capable of defining any ground-truth partition[37,40]. In the context of scRNA-seq studies, pairwise constraints can be constructed based on the cell distance computed using marker genes, cell sorting using flow cytometry, or other methods depending on real application scenarios.

To evaluate the performance of pairwise constraints, we applied our scDCC model to four scRNA-seq datasets generated from different sequencing platforms (Table 1, see data description in the methods part). We selected 10% of cells with known labels to generate constraints in each dataset and evaluated the performance of scDCC on the remaining 90% of cells. We show

that the prior information encoded as soft constraints could help inform the latent representations of the remaining cells and therefore improve the clustering performance. Three clustering metrics: normalized mutual information (NMI), clustering accuracy (CA), and adjusted Rand index (ARI) were applied to evaluate the performance from different aspects for each competing method. Specifically, the ranges of NMI and CA are from 0 to 1, while ARI can be negative. A larger value indicates better concordance between the predicted labels and ground truth. The number of pairwise constraints fed into the model explicitly controls how much prior information is applied in the clustering process. In the experiments, we varied the number of pairs from 0 to 6000, which represent a small fraction of all possible pairs (from 3.7 to 15.9% of all possible pairs in the selected 10% of cells among four datasets). We repeated the experiment for each setting ten times. As shown in Fig. 2, the clustering performance improves consistently across various datasets when the scDCC model takes more prior constraint information into account. For datasets that are difficult to cluster (e.g., worm neuron cells), imposing a small set of pairwise constraints significantly improves the results. With 6000 pairwise constraints, scDCC achieves acceptable performance on all four datasets (most clustering metrics >0.8), regardless of the baseline performances. To illustrate the contribution of the added constraints, we use t-SNE[4,5] to show the embedded representations of cells learned by different methods in the two-dimensional space (Fig. 3). A random subset of corresponding ML (blue lines) and CL (red lines) constraints are also plotted (Fig. 3). As shown, the latent representations learned by the ZINB model-based autoencoder are noisy and different labels are mixed. Although the representations from scDeepCluster could separate different clusters, the inconsistency against the constraints still exists. Finally, by incorporating the soft constraints into the model training, scDCC was able to precisely separate the clusters and the results are consistent with both ML (blue lines) and CL (red lines) constraints. Overall, these results show that pairwise constraints can help to learn a better representation during the end-to-end learning procedure and improve clustering performance.

The state-of-the-art clustering methods for scRNA-seq data, including CIDR[14], DCA[20] + k-means, DEC[27], MPSSC[16], PCA + k-means, SCVI[22] + k-means, SCVIS[23] + k-means, SIMLR[15], SC3[7] and Seurat[41], were selected as competing methods. In addition, we

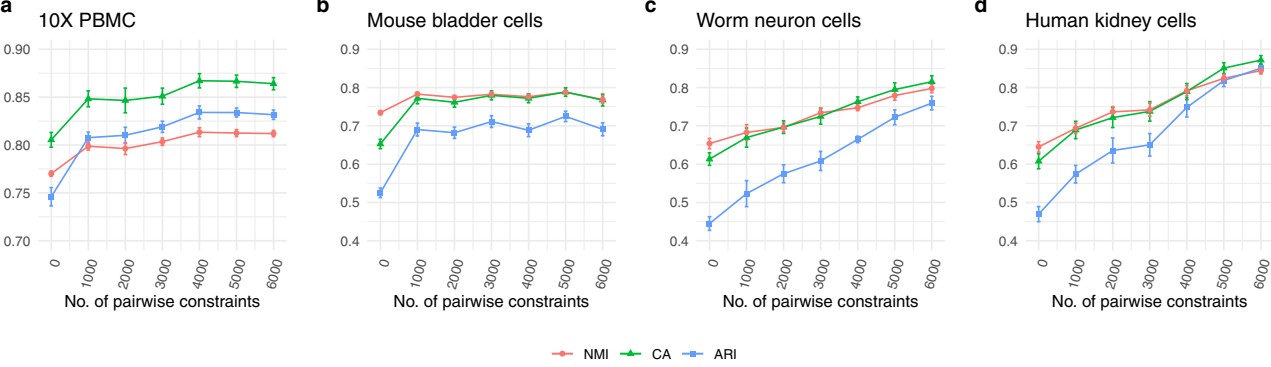

**Fig. 2 Performances of scDCC on small datasets. a** 10X PBMC; **b** Mouse bladder cells; **c** Worm neuron cells; **d** Human kidney cells. Clustering performances of scDCC on four small scRNA-seq datasets with different numbers of pairwise constraints, measured by NMI, CA, and ARI. All experiments are repeated ten times, and the means and standard errors are displayed.

also compared the proposed scDCC model with traditional constrained clustering algorithms (COP K-means[35] and MPC K-means[40]). Since some competing methods could not handle large-scale datasets, we randomly sampled 2100 cells from each dataset to form the final experimental datasets. The down-sampling procedure did not drop any group in any dataset. Following the same procedure, 10% of cells with labels were randomly selected to generate constraints, and all methods were evaluated and compared on the remaining 90% of cells. We note that competing methods, including CIDR, DCA + k-means, DEC, MPSSC, PCA + k-means, SCVI + k-means, SCVIS + k-means, SIMLR, SC3 and Seurat, are unable to utilize prior information. As a result, we first compared scDCC with these methods when no prior information is utilized (i.e., with 0 pairwise constraints). Of note, the proposed scDCC reduces to the scDeepCluster when no constraint information is used. For the randomly selected 2100 cells in each dataset, we observed that scDCC with 0 constraint outperformed most competing scRNA-seq clustering methods (some strong methods outperformed scDCC with 0 constraints on some datasets, such as SC3 and Seurat on mouse bladder cells), and by incorporating prior information, scDCC could perform significantly better than all competing methods (Figs. S1, S3, S5 and S7: left panels). For the full datasets, the performance between scDCC with 0 constraint and strong competing methods (e.g., SC3 and Seurat) was comparable, while considering the scDCC with constraint information, demonstrated significantly better performance than the competing methods as shown in Figs. S2, S4, S6, S8 (left panels). For different datasets, we observed consistent results that scDCC could deliver much better results than the traditional constrained clustering algorithms (COP K-means[36] and MPC K-means[42]) on scRNA-seq datasets (Figs. S1, S3, S5 and S7: left panels).

In real applications, we recognize that constraint information may not be 100% accurate (e.g., must-link cells may be erroneously labeled as cannot-link, and vice versa). To evaluate the robustness of the proposed method, we applied scDCC to the datasets with 5% and 10% erroneous pairwise constraints (Fig. S9). Though the errors in the constraints could degrade the performance, scDCC still achieves better clustering results by imposing constraints with 5% errors in all datasets (compared to scDCC with 0 constraint), indicating that scDCC is robust to noisy constraints. When the error rate increased to 10%, scDCC began to underperform in some datasets (e.g., mouse bladder cells) when the number of constraints increased. Therefore, users should take caution when adding highly erroneous constraints

With ongoing development of the sequencing technology, it is interesting to see the performance of scDCC on the very large scRNA-seq datasets. To this end, we applied scDCC to two

additional large datasets with 14,653 and 27,499 cells (Table 2). Again, we randomly selected 10% of cells with known labels to generate constraints and evaluated the performance of the remaining cells. We observed consistent improvements over baselines with the increasing number of constraints (Fig. 4, Fig. S10). As illustrated in Fig. S11, scDCC, when integrated with soft pairwise constraints, is robust to noise on the two large scRNA-seq datasets.

**Robustness on highly dispersed genes**. Gene filtering is widely applied in many single-cell analysis pipelines (e.g., Seurat[41]). One typical gene filtering strategy is to filter out low variable genes and only keep highly dispersed genes. Selecting highly dispersed genes could amplify the differences among cells but lose key information between cell clusters. To evaluate the robustness of scDCC on highly dispersed genes, we conducted experiments on the top 2000 highly dispersed genes of the four datasets and displayed the performances of scDCC and baseline methods in Figs. S1–8 (right panels). As we can see in Figs. S1–8 (right panels), scDCC without any constraints could provide comparable results to strong baseline methods in different datasets. When incorporating constraint information, scDCC was consistently better than other methods in both down-sampled and full data sets (Figs. S1–8: right panels). Figure S12 summarizes the detailed performance of scDCC in the four datasets (with 2100 selected cells or all cells) and different settings (all genes or top 2000 highly variable genes). As we increase the number of pairwise constraints constructed upon the randomly selected 10% of cells, the scDCC tends to provide better clustering performance on the remaining 90% of cells, which shows from another viewpoint that the prior information could be leveraged to obtain better clusters.

**Real applications and use cases**. In the previous sections, we mainly constructed the pairwise constraints for a small set of cells whose ground truth information is available (e.g., cell labels). In practice, depending on the concrete application and use case, we may generate constraints for the utilization of various information sources. Generating accurate constraints is the key to successfully apply the proposed scDCC algorithm to obtain robust and desired clustering results. Here we conducted two case studies to illustrate two different methods for encoding external information as constraints.

**Protein marker-based constraints**. The CITE-seq can profile expression levels of mRNAs and cell surface proteins simultaneously. Protein levels can provide additional information for cell type identification. We conducted the experiment on a CITE-seq

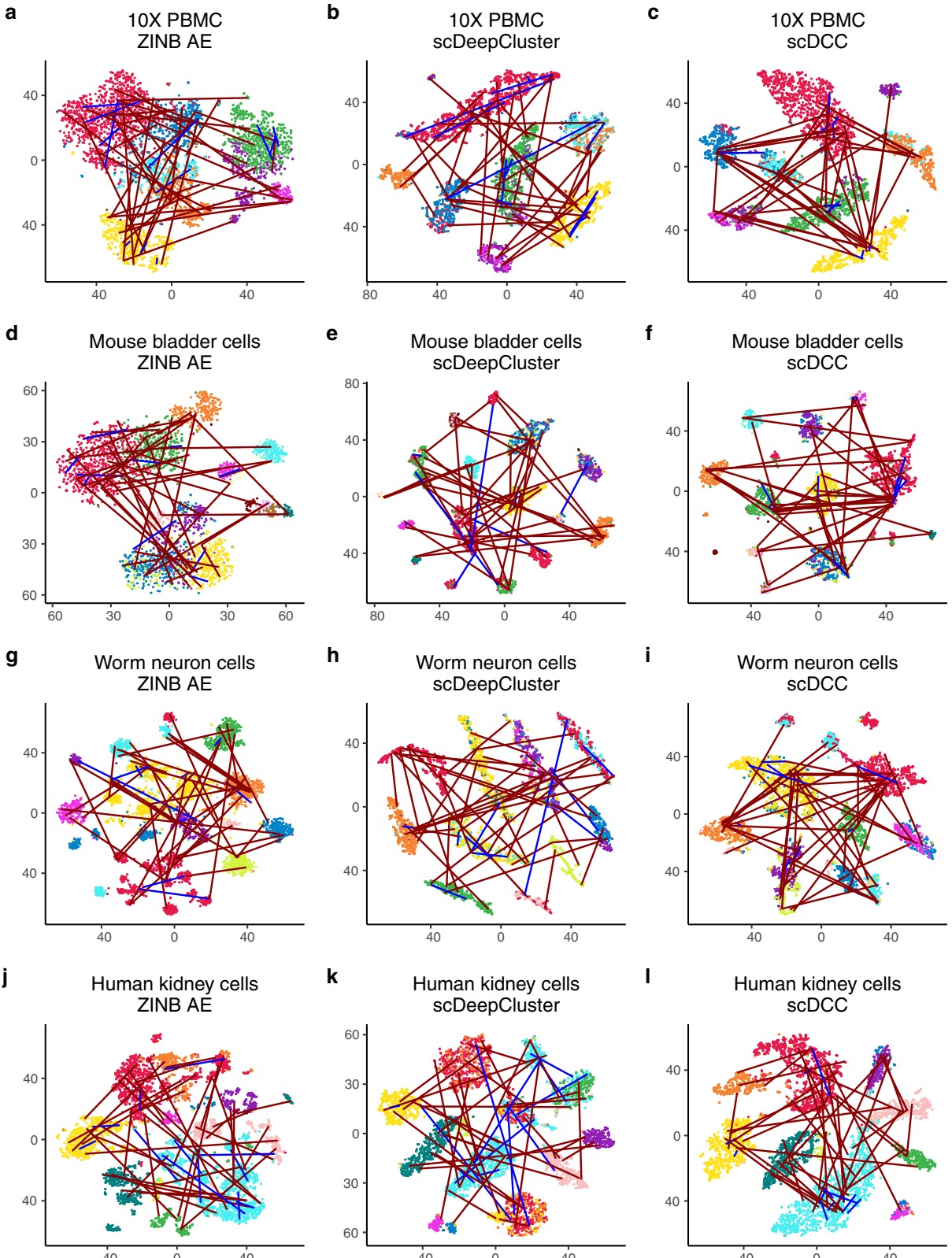

**Fig. 3 Latent representation visualization.** Comparison of 2D visualization of embedded representations of ZINB model-based autoencoder (**a**, **d**, **g**, **j**), scDeepCluster (**b**, **e**, **h**, **k**) and scDCC with pairwise constraints (**c**, **f**, **i**, **l**). The same instances and constraints are visualized for each dataset (**a**–**c**, 10X PBMC; **d**–**f** Mouse bladder cells; **g**–**i** Worm neuron cells; **j**–**l** Human kidney cells). The red lines indicate cannot-link and blue lines indicate must-link. The axes are arbitrary units. Each point represents a cell. The distinct colors of the points represent the true labels, and colors are arbitrarily selected.

PBMC dataset[43] with constraints built upon protein data. Specifically, to leverage the protein information and improve clustering on mRNAs, we generated constraints based on protein expression levels. We used a stringent criterion to generate ML and CL constraints (See "Methods": Obtain constraints). As shown in Fig. 5a, scDCC with 25,000 constraints significantly outperforms competing methods (including scDCC without any constraints) and achieves better results than SC3 (performed on the mRNA data), PhenoGraph[44], and k-means (performed on protein levels). We visualize the CD4 and CD8 protein levels in the clustering results of scDCC with and without constraints. Clusters were identified by differential expression analysis of CD4 and CD8 genes via the Wilcoxon test implemented in Seurat. Clusters with the most significantly high expressed CD4 and CD8 mRNAs were labeled as CD4 and CD8 T cells, respectively. As shown in Fig. 5b, in the clustering results of scDCC without constraint, CD4+/CD8− cells identified by the protein levels were erroneously labeled as CD8 T cells; in the clustering results of scDCC with constraints, protein levels of CD4 and CD8 were consistent with the clustering labels on mRNAs.

**Marker gene-based constraints**. After unsupervised clustering, cell-type marker genes are usually leveraged to annotate scRNA-seq data into predefined or de novo cell types. The success of such manual interpretations requires purified clusters with different marker genes highly expressed and concentrated only in some specific clusters. We show that the scDCC model with marker gene-based constraints could generate more interpretable clustering results.

Firstly, we conducted simulation experiments to systematically show that marker gene-based constraints can guide the model to separate the marker genes better with the increasing number of

constraints (Fig. S13, Supplementary Notes). Then we conducted real data experiments based on a human liver dataset[45], which had about 8000 cells. The authors provided clustering analysis results and annotated the clusters based on overexpression of marker genes. We used the marker genes to generate low dimensional latent representations for the downloaded 8000 cells. A t-SNE plot of all cells based on ZIFA latent features of marker genes was shown in Fig. S14. We construct constraints based on learned latent representations (See "Methods": Obtain constraints). We observed that by incorporating the constraint information, scDCC improved clustering performance significantly (Fig. 6a). It is noted that scDCC achieves better performance than PhenoGraph and k-means on the ZIFA latent features (scDCC with constraints vs k-means on ZIFA latent space, averaged measures: NMI: 0.905 vs 0.833, CA: 0.928 vs 0.928, ARI: 0.952 vs 0.924).

We show that scDCC with marker gene-based constraints could produce clusters with highly expressed marker genes concentrated in some specific clusters. We examined the distributions of these marker genes in each cluster. A gene is designated as a marker gene generally because it could represent a cell type. Ideally, biologists expect to see cells of the same type clustered together. As a result, it makes more biological sense if marker genes overexpress and concentrate on one cluster, but it is hard to interpret if a marker gene is highly expressed all over many clusters. Maintaining such cluster specificity of marker genes will bring enhanced biological interpretability. We defined a cluster specificity score for 55 marker genes, each of which is assigned to only one cell type. We expect each of them to be highly expressed in only one cluster. We apply DESeq2[46] to conduct differential expression analysis of these marker genes by comparing one cluster versus the others. The statistics reported by DESeq2 reflect the significance of the difference between the cluster and the rest. Each gene will then have one testing statistic for one cluster. The larger the testing statistic, the more significantly differentially expressed the gene clusters are. To summarize the level of a marker gene concentrated in one cluster (highly expressed in only one cluster), we define the maximum of the statistics of all clusters as the specificity score for a marker gene. We show that scDCC can increase cluster specificity of marker genes. We compared the averaged specificity scores of the ten repeats before and after adding constraints. As shown in Fig. 6b, adding constraints increases the specificity scores for

| Table 2 Summary of two large scRNA-seq datasets. | | | | |
|---|---|---|---|---|
| Dataset | Sequencing platform | Sample size /#Cell | #Genes | #Groups |
| Macosko mouse retina cells | Drop-seq | 14,653 | 11,422 | 39 |
| Shekhar mouse retina cells | Drop-seq | 27,499 | 13,166 | 19 |

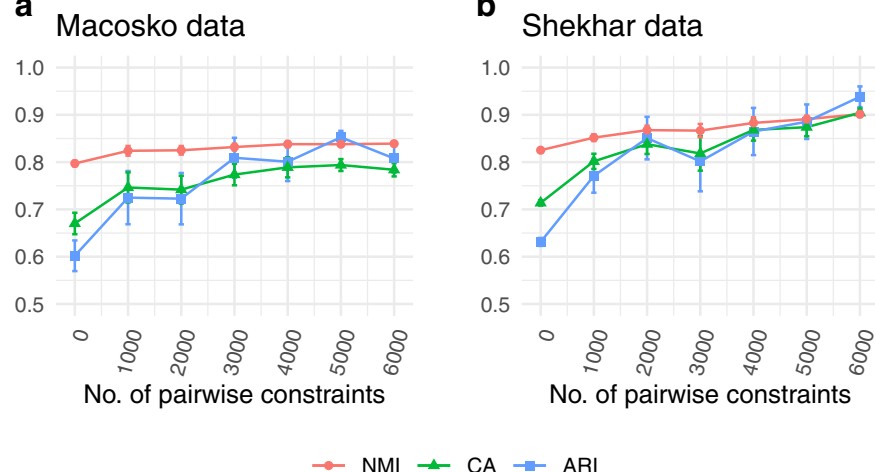

**Fig. 4 Performances of scDCC on large datasets. a** Macosko mouse retina cells; **b** Shekhar mouse retina cells. Clustering performances of scDCC on two large scRNA-seq datasets with different numbers of pairwise constraints, measured by NMI, CA, and ARI. All experiments are repeated ten times, and the means and standard errors are displayed.

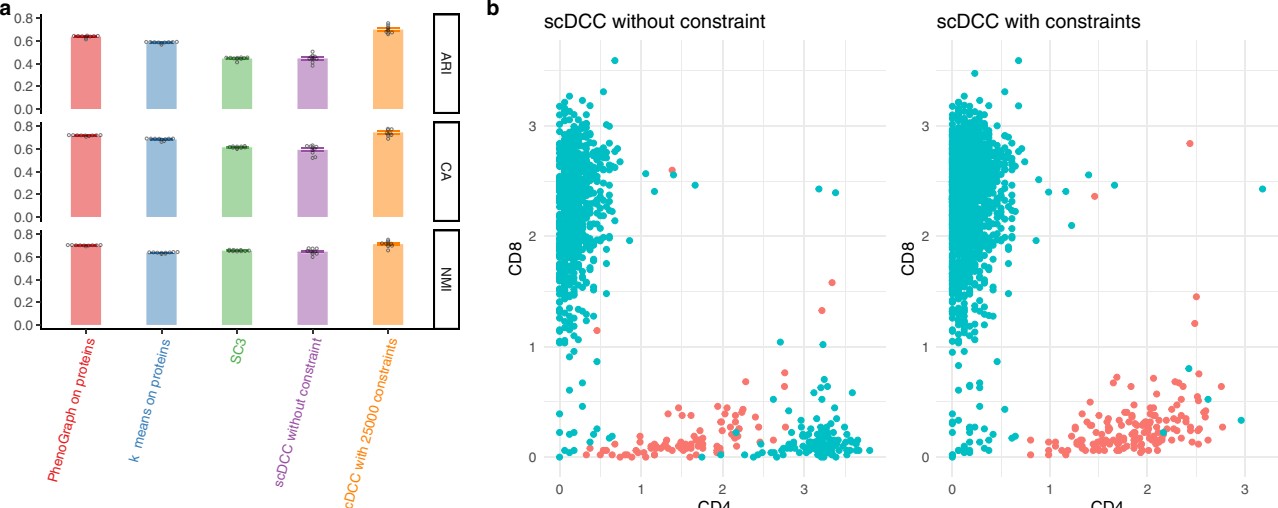

**Fig. 5 Clustering analysis on the CITE-seq PBMC data with protein-based constraints. a** Clustering performances of PhenoGraph and k-means on proteins, SC3, and scDCC (without and with constraints) on mRNAs of CITE-seq PBMC dataset, measured by NMI, CA, and ARI. All experiments are repeated ten times (one dot represents one experiment), and the means and standard errors are displayed. Constraints were generated from protein expression levels. **b** CD4 and CD8 protein expression levels in the identified CD4 and CD8 specific cells. Colors (Cyan represents CD8 cells and red represents CD4 cells) are cluster labels identified by scDCC with and without constraints on proteins. Cell labels were annotated by differential expression analysis.

most marker genes. Most points are above the diagonal line, and the improvement is significant ($p$-value = 0.00024, one-sided Wilcoxon test). The detailed specificity scores of each marker gene under the settings with and without constraints are summarized in Fig. S15.

Finally, we show that the cells with the same highly expressed marker genes tend to be clustered together when the scDCC model is trained with constraint information. We generated the t-SNE plots based on latent representations of scDCC without any constraints (Fig. 6c), and with 25,000 pairwise constraints (Fig. 6d). The cells with highly expressed marker genes are marked in red. As shown in Fig. 6c, d, cells with highly expressed marker genes distribute across different clusters when scDCC is trained without constraints. In contrast, they become more concentrated in one cluster when scDCC utilizes constraint information. The observation is consistent among marker genes AIF1, CD68, LYZ, and HAMP.

In summary, we addressed the problem of integrating prior knowledge into deep embedding clustering analysis for the scRNA-seq data and proposed a model-based constrained deep embedding clustering framework. Integrating soft constraints into the modeling process is flexible and can be applied to various real experimental settings. Our results on various small and large scRNA-seq datasets illustrate that even a small number of constraints can significantly improve the clustering performance. This observation is consistent with the expectation that constraints, which encode prior knowledge, help to learn better latent representations and clusters. Following most clustering studies, we assume the number of clusters $k$ has already been defined in all experiments. In practice, this information is usually unknown. We propose the use of an elbow method[26] or a density method[47] to estimate the value of $k$. We propose one simple distance-based method and one clustering-based method for integrating prior information. The key is to generate accurate constraints so the defined must-links and cannot-links can faithfully represent domain knowledge. We hope the two methods could ignite the creativity of users to construct constraints for encoding their own domain knowledge. As future

work, we will explore different kinds of domain information and develop general ways of constructing constraints from implicit knowledge.

## Methods

**Read count data preprocessing and transformation**. Following the methods of Tian et al.[26], we applied the Python package SCANPY[48] (version 1.4.4) to preprocess the raw scRNA-seq read count data. Firstly, we filter out genes with no count in any cell. Secondly, we calculate the size factors for each cell and normalize the read counts by the library size, such that the total counts are the same across cells. Formally, let's denote the library size (i.e., the number of total read counts) of cell $i$ as $s_i$; the size factor of cell $i$ is then $s_i$/median($s$). Finally, we take the log transformation and scale the read counts to have unit variance and zero mean. The transformed read count matrix is used as the input for our denoising ZINB model-based autoencoder. When calculating the ZINB loss, we use the raw count matrix[20,22,26].

**Denoising ZINB model-based autoencoder**. The autoencoder is a special artificial neural network with a low-dimensional bottleneck layer capable of learning efficient nonlinear representations in an unsupervised manner[21]. Among various autoencoder models, the denoising autoencoder receives corrupted data (e.g., by adding Gaussian noises) as inputs and is trained to reconstruct the original uncorrupted data[49]. It has proven to be robust and powerful in learning a good representation from noisy datasets. Here, we apply the denoising autoencoder to map the preprocessed read counts to a low dimensional embedded space to carry out clustering. Formally, denoting the preprocessed input as $\bar{X}$, the input for denoising autoencoder is

$$\bar{X}^{\text{corrupt}} = \bar{X} + e \tag{1}$$

where $e$ is the Gaussian noise. We define encoder function as $Z = f_W(X^{\text{corrupt}})$ and decoder function $X' = g_{W'}(Z)$. Encoder and decoder functions are both multi-layered fully connected neural networks with the rectifier activation function (ReLU)[50]. Here $W$ and $W'$ are the learnable weights. The learning process of the denoising autoencoder is to minimize the loss function

$$L\left(X, g_{W'}\left(f_W\left(\bar{X}^{\text{corrupt}}\right)\right)\right) \tag{2}$$

with regard to the learnable weights.

Following previous studies[20,22,26], we employ a ZINB model-based autoencoder to model the scRNA-seq data. Unlike the traditional autoencoder methods, ZINB model-based autoencoder uses the likelihood of a ZINB distribution to characterize scRNA-seq count data. Let $X_{ij}^{\text{count}}$ be the read count for cell $i$ and gene $j$ in the scRNA-seq raw count matrix. The ZINB distribution is parameterized by a negative binomial distribution with mean $\mu_{ij}$ and dispersion $\theta_{ij}$, and an additional

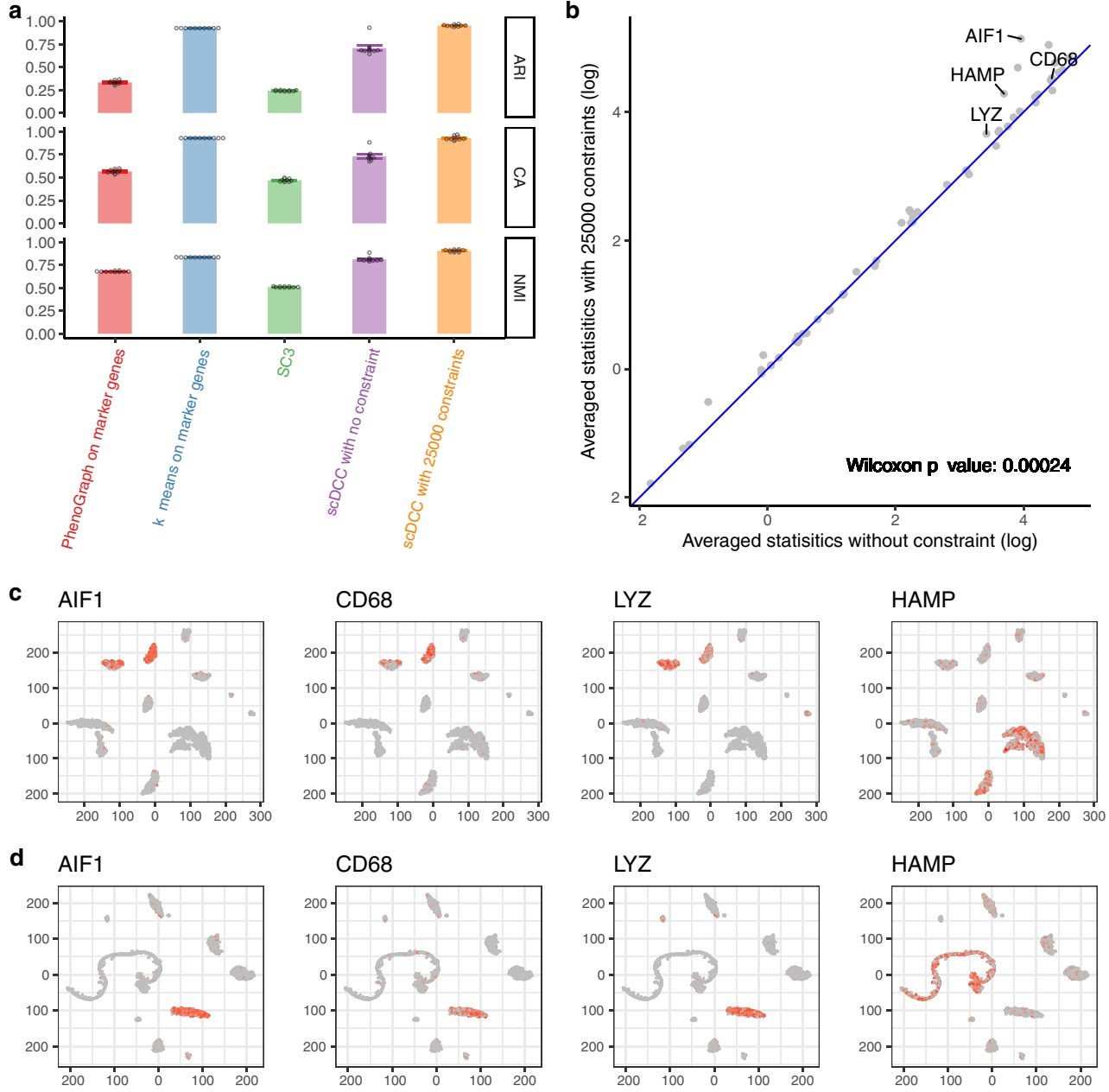

**Fig. 6 Clustering analysis on the human liver cells with marker gene-based constraints. a** Clustering performances of PhenoGraph and k-means on ZIFA representations of marker genes, SC3, and scDCC (with and without constraints) on mRNAs of the human liver dataset measured by NMI, CA, and ARI. All experiments are repeated ten times (one dot represents one experiment), and the means and standard errors are displayed. Constraints were constructed by marker genes. **b** Average (with constraints comparing to without constraints) of specificity scores of 55 marker genes. One-sided Wilcoxon test *p*-value is also displayed (with constraints vs. without constraints). **c** Marker gene expression (log normalized counts) for each cell marked on the t-SNE plot based on latent representations of scDCC without any constraint. **d** Marker gene expression (log normalized counts) for each cell marked on the t-SNE plot based on latent representations of scDCC with 25,000 pairwise constraints. **c**, **d** t-SNE plots calculated from the bottleneck features of scDCC with and without constraints, respectively. Colors represent the relative expression levels; gray and red represent low and high expression levels, respectively. Axes are arbitrary values.

coefficient $\pi_{ij}$ that represents the probability of dropout events:

$$\text{NB}\left(X_{ij}^{\text{count}}\Big|\mu_{ij}, \theta_{ij}\right) = \frac{\Gamma\left(X_{ij}^{\text{count}} + \theta_{ij}\right)}{X_{ij}^{\text{count}}!\,\Gamma\left(\theta_{ij}\right)}\left(\frac{\theta_{ij}}{\theta_{ij} + \mu_{ij}}\right)^{\theta_{ij}}\left(\frac{\mu_{ij}}{\theta_{ij} + \mu_{ij}}\right)^{X_{ij}^{\text{count}}} \quad (3)$$

$$\text{ZINB}\left(X_{ij}^{\text{count}}\Big|\pi_{ij}, \mu_{ij}, \theta_{ij}\right) = \pi_{ij}\delta_0\left(X_{ij}^{\text{count}}\right) + \left(1 - \pi_{ij}\right)\text{NB}\left(X_{ij}^{\text{count}}\Big|\mu_{ij}, \theta_{ij}\right) \quad (4)$$

Letting $D = g'_{W'}\left(f_W\left(\bar{X}^{\text{corrupt}}\right)\right)$ be the output matrix of the last hidden layer of decoder, we append three independent fully connected layers to $D$ to estimate the

ZINB parameters:

$$\begin{aligned}
M &= \text{diag}(s_i) \times \exp\left(W_\mu D\right) \\
\Theta &= \exp\left(W_\theta D\right) \\
\Pi &= \text{sigmoid}\left(W_\pi D\right)
\end{aligned} \quad (5)$$

where $M$, $\Theta$, and $\Pi$ represent the matrix form of the estimated mean, dispersion, and dropout probability, respectively. The size factors $s_i$ are precalculated (See Section: Read count data preprocessing and transformation) and included as an independent input to the ZINB model-based autoencoder. The loss function of ZINB model-based autoencoder is the sum of the negative log of ZINB likelihood

of each data entry

$$L_{\text{ZINB}} = \sum_{ij} -\log\left(\text{ZINB}\left(X_{ij}^{\text{count}}\Big|\pi_{ij}, \mu_{ij}, \theta_{ij}\right)\right) \quad (6)$$

**ZINB model-based deep embedded clustering.** Clustering analysis is conducted on the embedded latent space[27,28]. Let $X$ denote a set of $n$ cells with $x_i \in \mathbb{N}^d$ representing the read counts of $d$ genes in the $I$ th cell. scDCC applies the denoising ZINB model-based autoencoder to learn a non-linear mapping $f_W : x_i \to z_i$ and transforms the input $X$ to a low-dimensional latent space $Z$. Let $Q$ be the distribution of soft labels measured by Student's $t$-distribution and $P$ be the derived target distribution from $Q$. We define the clustering loss function as the Kullback-Leibler (KL) divergence between $P$ and $Q$

$$L_c = KL(P \parallel Q) = \sum_i \sum_j p_{ij} \log\frac{p_{ij}}{q_{ij}} \quad (7)$$

where $q_{ij}$ is the soft label of embedded point $z_i$. Specifically,

$$q_{ij} = \frac{\left(1 + \|z_i - \mu_j\|^2\right)^{-1}}{\sum_{j'}\left(1 + \|z_i - \mu_{j'}\|^2\right)^{-1}} \quad (8)$$

measures the similarity between point $z_i$ and cluster center $\mu_j$ calculated by the Student's $t$-distribution[4,5] and

$$p_{ij} = \frac{q_{ij}^2/\sum_i q_{ij}}{\sum_{j'}\left(q_{ij'}^2/\sum_i q_{ij'}\right)} \quad (9)$$

represents the target distribution applied in the self-training[51]. At each iteration, minimizing the loss function $L_c$ will push $Q$ moving towards the derived target distribution $P$.

**Pairwise constraints.** Pairwise constraint can have two types: must-link (ML) and cannot-link (CL) [38]. The loss of must-link constraint forces the two instances to have similar soft labels:

$$L_{\text{ML}} = -\sum_{(a,b)\in \text{ML}} \log \sum_j q_{aj} \times q_{bj} \quad (10)$$

In contrast, the loss of cannot-link encourages different soft labels:

$$L_{\text{CL}} = -\sum_{(a,b)\in \text{CL}} \log(1 - \sum_j q_{aj} \times q_{bj}) \quad (11)$$

**Deep constrained clustering framework.** We first pre-train the denoising ZINB model-based autoencoder to minimize the reconstruction loss ($L_{\text{ZINB}}$). We initialize the clustering centroids by performing standard k-means clustering on the learned embedded latent vectors. We then jointly optimize losses

$$L = L_{\text{ZINB}} + \gamma L_{\text{clustering}} + \gamma' L_{\text{constraint}} \quad (12)$$

where $L_{\text{ZINB}}$, $L_{\text{clustering}}$ and $L_{\text{constraint}}$ are the ZINB reconstruction loss, the clustering loss, and the constraint loss, respectively; $\gamma$ and $\gamma' > 0$ control the relative weights of the two losses. Combining the ZINB loss and clustering loss can preserve the local structure of the data generating distribution[28]. During the clustering stage, we optimize the ZINB loss and clustering loss per batch of data points, and optimize the constraint loss per batch of constraints.

**Implementation.** scDCC is implemented in Python 3 (version 3.7.6) using PyTorch[52] (version 1.5). The sizes of hidden layers in ZINB model-based autoencoder are set to be (256, 64, 32, 64, 256), where the bottleneck layer's size is 32. The standard deviation of Gaussian random noise is 2.5. Adam with AMSGrad variant[53,54] and Adadelta[55] are applied for pretraining stage and clustering stage, respectively. The parameters of Adam optimizer are set with initial learning rate $lr = 0.001$, $\beta_1 = 0.9$, and $\beta_2 = 0.999$, and parameters of Adadelta optimizer are set to be of $lr = 1.0$ and $rho = 0.95$. The choice of $\gamma$ follows scDeepCluster's setting of 1. The weight of constraint loss $\gamma'$ is set to be 1 for all experiments. The batch size for pretraining and clustering is 256. We pretrained the autoencoder 300 epochs. The convergence threshold for clustering stage is 0.1% of the changed clustering labels per epoch. All experiments are conducted on Nvidia Tesla P100 (16 G) GPU.

**Constraint construction.** For the datasets listed in Table 1 and Table 2, we randomly selected 10% of the total cells as a hold-out cell set to generate pairwise constraints and left the remaining cells for evaluation. Specifically, we randomly selected 1000, 2000, 3000, 4000, 5000, and 6000 pairs of cells from the hold-out set and defined must-link and cannot-link constraints based on the label information we collected. Then, we ran scDCC on the whole cells with the generated constraints and evaluated the performance on the remaining 90% of cells.

The CITE-seq PBMC dataset provides read counts of both mRNAs and proteins. Firstly, protein counts were normalized and scaled by the Seurat "NormalizeData" function with the setting of "CLR". Secondly, we calculated

Euclidean distances for all possible pairs of cells based on the normalized protein data and chose the 0.5th and 95th percentile of all pairwise distances as the must-link and cannot-link constraint cutoffs. Specifically, we repeatedly sampled two cells, and if the Euclidean distance between the two cells was less than the 0.5th percentile of all pairwise distances, we defined it as a must-link constraint; if the Euclidean distance between the two cells was greater than the 95th percentile of all pairwise distances, we defined it as a cannot-link constraint. We generated 20,000 constraints based on all protein levels. To separate CD4 and CD8 T cells, we further added 5000 constraints based on following rules: if one cell has high CD4 protein level (>70th percentile) and low CD8 protein level (<30th percentile) and another cell has high CD8 protein level (>70th percentile) and low CD4 protein level (<30th percentile), then a cannot-link is constructed. To further identify subtypes of CD4 and CD8 T cells (CD8+CD27−, CD8+CD27+, CD4+CD27+, CD4+CD27−DR+, CD4+CD27−DR−), we generate must-links for each subtype. Taking the CD8+CD27+ T cells as an example, we require that the two randomly selected cells to form a must-link constraint should have both high CD8 protein levels (>85th percentile) and high CD27 protein levels (>85th percentile). In contrast, the two cells to form a must-link constraint for the CD8+CD27− subtype should have high CD8 protein levels (>85th percentile) but low CD27 protein levels (<30th percentile). For CD4+CD27+, CD4+CD27−DR+, CD4+CD27−DR− cells, we applied similar rules to construct must-links.

In the Human liver dataset, we used marker genes to generate constraints. The table of (revised) marker genes was downloaded from ref. [35]. We first used "NormalizeData" function from Seurat package[41] to normalize the raw count matrix and obtained normalized counts of the 63 marker genes, among which 55 marker genes uniquely belonged to one cell type. We applied a zero-inflated factor analysis (ZIFA) method[56] to reduce the dimensions of the marker gene matrix to 10 (Fig. S14). Constraints were generated based on the ZIFA latent representations. Specifically, we applied k-means on the ZIFA latent representations, and used k-means results as the pseudo labels. Must-link and cannot-link constraints were defined on these k-means labels. After obtaining clustering results, we applied DESeq2[46] to compare levels of differential expression of the 55 marker genes. Dispersions were estimated using "mean" for the *fitType* parameter. We defined the level of differential expression by the Wald statistics reported by DESeq2.

**Competing methods.** CIDR[14] (https://github.com/VCCRI/CIDR), DCA[20] (https://github.com/theislab/dca), DEC[27] (https://github.com/XifengGuo/DEC-keras), MPSSC[16] (https://github.com/ishspsy/project/tree/master/MPSSC), PCA + k-means, scDCC (without constraint, https://github.com/ttgump/scDCC), SCVI[22] (https://github.com/YosefLab/scVI), SCVIS[23] (https://github.com/shahcompbio/scvis), SIMLR[15] (https://bioconductor.org/packages/release/bioc/html/SIMLR.html), SC3[7] (https://bioconductor.org/packages/release/bioc/html/SC3.html), Seurat[41] (http://satijalab.org/seurat/), COP K-means[36] (R package "conclust") and MPC K-means[42] (R package "conclust") are used as competing methods. Packages and APIs developed by original authors are applied to conduct the experiments, when available. In addition, the raw count matrices are pre-processed based on the steps described in previous works for each competing method. Following Lin et al.[14], we construct scData R objects based on the raw count matrices and conduct a series of clustering steps: determining the dropout events and imputation weighting thresholds, computing the CIDR dissimilarity matrix, reducing the dimensionality, and clustering. We apply DCA to denoise and impute the read counts data. Principal component analysis (PCA) is applied to reduce the high-dimensional denoised read count matrix to the 2D space, and k-means clustering is conducted on the projected 2D space to predict the final labels. SCVI uses stochastic optimization and deep neural networks to aggregate information across similar cells and genes and learn a probabilistic representation to approximate the distributions that underlie observed expression values. Following Lopez et al.[22], we retained the top 700 genes ordered by variance computed on the log gene expressions. SCVIS is a variational autoencoder based model and could capture the low-dimensional representation of scRNA-seq data. Following Ding et al.[23], the gene expression is quantified as log2(CPM/10 + 1), where 'CPM' stands for 'counts per million', and the pre-processed matrix is then projected into a 100-dimensional space via PCA for the SCVIS analysis. As DCA, SCVI, and SCVIS mainly focus on imputation or learning good representations, k-means clustering is performed on their results to obtain the final clustering labels. Therefore, we denote them as 'DCA + k-means', 'SCVI + k-means', and 'SCVIS + k-means', respectively. DEC and scDCC share the same input that the raw count matrix is library-size normalized, log transformed, scaled, and centered. Default settings and hyperparameters are selected for different methods according to their original publications and online user guides. For example, the parameters for MPSSC are rho = 0.2, lam = 0.0001, lam2 = 0.0001, eta = 1, and c = 0.1, and the sizes of the hidden layers for DEC are 500, 500, 2,000, and 10. For MPSSC, SIMLR, PCA + k-means, COP K-means, and MPC K-means, the read count matrix is normalized by library size and log-transformed. PCA + k-means, which applies PCA to project the processed raw read count matrix to 2D space directly, followed by k-means clustering is chosen as a baseline method for evaluating the impacts of linear and non-linear representations. It should be noted that traditional constrained clustering algorithms, COP K-means and MPC K-means, are also applied on the PCA projected 2D space. SC3 first calculates three different distances- Euclidean, Pearson and Spearman metrics- to construct distance matrices. Then, a consensus spectral

clustering combines all distance matrices to archive high-accurate and robust results. Seurat is developed and maintained by the Satija lab, which is an R package integrated with state-of-the-art methods and has been broadly applied by biological researchers for QC, analysis, and exploration of single-cell RNA-seq data. Seurat identifies clusters of cells by a SNN modularity optimization-based clustering algorithm. Parameters for Seurat are set to be default (e.g., resolution = 0.8). PhenoGraph[44] (https://github.com/jacoblevine/PhenoGraph) is a clustering method designed for high-dimensional single-cell data. It works by creating a graph representing phenotypic similarities between cells and identifying communities in this graph. For both Seurat and PhenoGraph, we selected the default settings with Louvain algorithm as their core implementations.

**Evaluation metrics**. NMI[57], CA[27] and ARI[58] are used as metrics to compare the performance of different methods.

Let $U = \{U_1, U_2, \ldots, U_{C_U}\}$ and $V = \{V_1, V_2, \ldots, V_{C_V}\}$ be the predicted and ground-truth clusters on a set of $n$ data points. NMI is defined as follows:

$$\mathrm{NMI} = \frac{I(U, V)}{\max\{H(U), H(V)\}} \quad (13)$$

where $I(U, V) = \sum_{p=1}^{C_U} \sum_{q=1}^{C_V} |U_p \cap V_q| \log \frac{n|U_p \cap V_q|}{|U_p| \times |V_q|}$ represents the mutual information between $U$ and $V$; $H(U) = -\sum_{p=1}^{C_U} |U_p| \log \frac{|U_p|}{n}$ and $H(V) = -\sum_{q=1}^{C_V} |V_q| \log \frac{|V_q|}{n}$ are the entropies.

CA is defined as the best matching between the predicted clusters and the ground-truth clusters. Let $l_i$ and $\hat{l}_i$ be the ground-truth label and the prediction of the clustering algorithm for the data point $i$. The CA is calculated as follows:

$$CA = \max_m \sum_{i=1}^{n} \mathbb{1} \frac{\left\{ l_i = m(\hat{l}_i) \right\}}{n} \quad (14)$$

where $n$ is the number of data points and $m$ ranges over all possible one-to-one mapping between cluster assignments and true labels. The best mapping can be efficiently searched by the Hungarian algorithm[59].

In data clustering, Rand Index[60] measures the agreement/similarity between cluster assignments $U$ and $V$. The Adjust Rand Index (ARI) is the corrected-for-chance version of the Rand index[58]. Assume that $a$ is the number of pairs of two objects in the same group in both $U$ and $V$; $b$ is the number of pairs of two objects in different groups in both $U$ and $V$; $c$ is the number of pairs of two objects in the same group in $U$ but in different groups in $V$; and $d$ is the number of pairs of two objects in different groups in $U$ but in the same group in $V$. The ARI is formally defined as

$$\mathrm{ARI} = \frac{\binom{n}{2}(a+d) - [(a+b)(a+c) + (c+d)(b+d)]}{\binom{n}{2} - [(a+b)(a+c) + (c+d)(b+d)]}. \quad (15)$$

**Real scRNA-seq datasets**. 10X PBMC dataset was provided by the 10X scRNA-seq platform[61], which profiled the transcriptome of about 4000 peripheral blood mononuclear cells (PBMCs) from a healthy donor. The 10X PBMC dataset was downloaded from the website of 10X genomics (https://support.10xgenomics.com/single-cell-gene-expression/datasets/2.1.0/pbmc4k). We downloaded the filtered gene/cell matrix. Cell labels identified by graph-based clustering (https://support.10xgenomics.com/single-cell-gene-expression/software/pipelines/latest/output/analysis) were used as ground-truth labels.

The mouse bladder cells dataset was provided by the Mouse Cell Atlas project[62] (https://figshare.com/s/865e694ad06d5857db4b). We downloaded the count matrix of all 400,000 single cells sorted by tissues. We used the cell types annotated by the authors[61]. From the raw count matrix, we selected the cells from bladder tissue for the experiments.

The worm neuron cells dataset was profiled by the sci-RNA-seq platform (single-cell combinatorial indexing RNA sequencing)[63]. They profiled about 50,000 cells from the nematode *Caenorhabditis elegans* at the L2 larval stage and identified the cell types (http://atlas.gs.washington.edu/worm-rna/docs/). We selected the subset of the neural cells and removed the cells with the label of "Unclassified neurons". As a result, we obtained 4186 neural cells for the experiments.

The human kidney dataset[64] was downloaded from https://github.com/xuebaliang/scziDesk/tree/master/dataset/Young. Authors profiled human renal tumors and normal tissue from fetal, pediatric, and adult kidneys. The dataset contains 5685 cells grouped into 11 clusters and each cell has 25,215 genes.

The Macosko mouse retina cells[65] and Shekhar mouse retina cells[66] datasets were profiled by the Drop-seq platform. Macosko mouse retina cells were downloaded from https://scrnaseq-public-datasets.s3.amazonaws.com/scater-objects/macosko.rds. Macosko et al. profiled more than 40,000 cells and identified cell labels via PCA and density-based clustering, and they further validated cell labels by differential gene expression analysis. We downloaded the Macosko dataset and filtered cells and genes. Concretely, cells with <700 genes and genes with <3 reads in 3 cells were filtered out. As a result, we obtained 14,653 cells by 11,422

genes among 39 clusters. Shekhar mouse retina cells were downloaded from https://scrnaseq-public-datasets.s3.amazonaws.com/scater-objects/shekhar.rds. Labels were provided by authors which we kept unchanged.

CITE-seq PBMC data[43] was downloaded from https://github.com/canzarlab/Specter. The dataset contains 3,762 cells, 18,677 genes and 49 protein markers. By clustering analysis and gene differential expression analysis, the dataset was divided into 12 clusters. We selected the top 2000 dispersed genes to conduct clustering experiments.

The human liver dataset was downloaded from https://github.com/BaderLab/scClustViz. The dataset contains 8,444 cells by 20,007 genes. The authors provided the clustering results and marker gene list. The true labels were obtained by using the "labelCellTypes" function from the scClustViz package[67]. As a result, the dataset contains 11 cell types. We selected the top 5000 dispersed genes to conduct clustering experiments.

**Reporting summary**. Further information on research design is available in the Nature Research Reporting Summary linked to this article.

## Data availability

The scRNA-seq datasets supporting this study are available publicly: 10X PBMC dataset (https://support.10xgenomics.com/single-cell-gene-expression/datasets/2.1.0/pbmc4k); mouse bladder cells (https://figshare.com/s/865e694ad06d5857db4b); worm neuron cells (http://atlas.gs.washington.edu/worm-rna/docs/); human kidney cells (https://github.com/xuebaliang/scziDesk/tree/master/dataset/Young); Macosko mouse retina cells (https://scrnaseq-public-datasets.s3.amazonaws.com/scater-objects/macosko.rds); Shekhar mouse retina cells (https://scrnaseq-public-datasets.s3.amazonaws.com/scater-objects/shekhar.rds); CITE-seq dataset (https://github.com/canzarlab/Specter/tree/master/data); human liver dataset (https://github.com/BaderLab/scClustViz). All datasets can be found on GitHub: https://github.com/ttgump/scDCC/tree/master/data.

## Code availability

The code that supports the results can be found on GitHub: https://github.com/ttgump/scDCC.

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

## Acknowledgements

This work was supported by Extreme Science and Engineering Discovery Environment (XSEDE) through allocation CIE160021 and CIE170034 (supported by National Science Foundation Grant No. ACI-1548562). We thank Akaash Patel for proofreading the manuscript which improved the clarity of the paper.

## Author contributions

Z.W. and H.H. conceived and supervised the project. T.T. and J.Z. designed the method and conducted the experiments. X.L. helped to conduct experiments for competing methods. T.T., J.Z., Z.W., and H.H. wrote the manuscript. All authors approved the manuscript.

## Competing interests

The authors declare no competing interests.
