## [Peer Review File · Nature Communications]

Reviewers' Comments:

Reviewer #1:

Remarks to the Author:

In this manuscript, Tian Tian et al. proposed a clustering method named scDCC to integrate biological domain knowledge into the clustering step of scRNA-seq data. They converted prior knowledge into pairwise and triplet constraints and integrated them as additional terms into the loss function for model optimization. However, the authors need to address some critical issues before the acceptance for publication:

1. It is not surprising that scDCC without constraint information, which reduces to their previously developed scDeepCluster, significantly outperforms baseline methods, because the adopted datasets and baseline methods in this work are almost the same as that of scDeepCluster. The authors should assess the performance using more scRNA-seq datasets or, at least, compare with other more widely used clustering methods, such as SC3 and Seurat. Besides, the graph-based Louvain clustering usually provides superior performance than k-means.

2. I believe that incorporating prior information could effectively improve the performance. However, more convincing evaluation should be performed to avoid information leakage. Briefly, the authors used cell labels to define ML and CL constraints, which means feeding true results to the model (which cell-pair should be clustered and which cell-pair be separated). Although 6000 constraints only represented about 0.27% of all possible pairs in 2100 cells, the constraints existed in a single-cell network and may contain the neighbor information of most cells. The clustering performance hence improves consistently across various datasets when the scDCC model takes more prior constraint information into account. It would be better if the authors compare the clustering performance for only the cells which do not exist in the constraints. In addition, the ML and CL constraints in the visualization in Figure 3 should be independent with the constraints used to train the scDCC model.

3. Analogously, triplet constraints were generated based on the embedding trained by scDCC with 20,000 pairwise constraints, which again indicates that the model obtained some true results in advance. It would be better if the authors pre-train scDCC without constraints to obtain the embedding.

4. It is good to see the authors evaluate the robustness of scDCC using 5% erroneous constraints. Given that the prior information may be very noisy in real application scenarios, I am interested in what percentage of erroneous constraints will invalidate this model?

5. The model with combined constraints seems to use twice as many constraints as the one with only pairwise or triplet constraints. If so, the result of the comparison in Figure 6 may not be fair.

6. In the example of marker gene-based constraints (Figure 7a), it would be better if the authors could compare the performance of Louvain clustering or k-means on the ZIFA latent representations of the marker gene matrix. Besides, it should be compared with the baseline methods.

7. Considering that the authors listed a few cell type-specific signature sets as prior knowledges, it would be better if they can discuss in more detail about how to integrate these prior knowledges.

Minor points:

1. <https://github.com/ttgump/scDCC> is unavailable.

2. In the last paragraph of Results, it should be 'with 6000 triplet constraints (Figure 7e)'.

3. The authors overemphasized that scDCC does not require exact label information, since this is a

clustering task where we seek to know which samples belong to the same cluster rather than a classification where we want to predict the exact label of each sample.

Reviewer #2:

Remarks to the Author:

This article introduces a deep neural network model for constrained clustering of single-cell RNAseq profiles. The constraints are based on log Bernoulli likelihood on connectivities between pairs or triples of samples. The methods are applied to several real single-cell RNAseq datasets. The paper is well-written and the methods are clearly described. There are some critical questions to address in the validation.

1. The clustering performance shown in Figure 2, 4, 5 and 6 are not properly evaluated since the training pairs are also included in the calculation of the measures. Similar to semi-supervised learning, the clustering of the constrained model can be very sensitive to any kind of supervised information. In principle, any samples that are presented with supervised information (such as pairwise relations in this model) should not be used in the evaluation. The proper experimental setup should be split the samples in the dataset into a training set and a test set. The constrained pairs/triples should only be sampled from the training set and the clustering results should only be reported for the samples in the test set. If there are a excessive number of hyper-parameters such as neural network structures, a validation set might also be needed. Given the model was only tested on a subset of each of the datasets, there should be enough samples for the train/test setup.

Without the correct validation, it is still unclear how much additional information is introduced by the pairs/triples to improve clustering in the current work.

2. The motivation of modeling the pairwise/triplewise relations in real studies is still unclear. The application to the five real datasets was taken by sampling from the final complete clustering in those studies, in which none of the dataset was originally generated with the pairwise knowledge. The common scenario is the case that a small set of cells are manually annotated and then used to train a model for classifying/clustering more single-cells from additional experiments, which is the case of supervised learning or semi-supervised learning. The pairwise constraint is still uncommon. It is important to justify the usefulness this model with real application scenarios, which is missing in the experiments or discussions.

A minor note:

3. On paper 9, the definition of p_{ij} abuses the use of indexes j and j' in the summations, which creates some confusion in understanding the statistics (which might be an error from the original equation in citation 46).

Reviewer #1 (Remarks to the Author):

In this manuscript, Tian Tian et al. proposed a clustering method named scDCC to integrate biological domain knowledge into the clustering step of scRNA-seq data. They converted prior knowledge into pairwise and triplet constraints and integrated them as additional terms into the loss function for model optimization. However, the authors need to address some critical issues before the acceptance for publication:

1. It is not surprising that scDCC without constraint information, which reduces to their previously developed scDeepCluster, significantly outperforms baseline methods, because the adopted datasets and baseline methods in this work are almost the same as that of scDeepCluster. The authors should assess the performance using more scRNA-seq datasets or, at least, compare with other more widely used clustering methods, such as SC3 and Seurat. Besides, the graph-based Louvain clustering usually provides superior performance than k-means.

Response

We thank the reviewer for raising these points. Following the reviewer's suggestion, we added a new human kidney dataset, and two additional state-of-the-art baseline methods: SC3¹ and Seurat² (We apply the Louvain algorithm (default) from the Seurat package to cluster the cells) for evaluation. As we can see in the revised Figures (Figure S1-S8, S10), without any constraints, scDCC could achieve comparable clustering performance as SC3 and Seurat (Louvain algorithm). To further illustrate the usage of our model, we added experiments based on a new "CITE-seq PBMC" dataset. In the two case studies of the manuscript (CITE-seq PBMC data and Human liver data), we generated constraints based on protein and marker gene data, respectively. To compare performance, in the revised version, we added a baseline method PhenoGraph which is based on Louvain clustering on protein and marker gene levels, respectively (Figure 5a and Figure 6a). We can see that by incorporating constraints, scDCC can outperform PhenoGraph³.

2. I believe that incorporating prior information could effectively improve the performance. However, more convincing evaluation should be performed to avoid information leakage. Briefly, the authors used cell labels to define ML and CL constraints, which means feeding true results to the model (which cell-pair should be clustered and which cell-pair be separated). Although 6000 constraints only represented about 0.27% of all possible pairs in 2100 cells, the constraints existed in a single-cell network and may contain the neighbor information of most cells. The clustering performance hence improves consistently across various datasets when the scDCC model takes more prior constraint information into account. It would be better if the authors compare the clustering performance for only the cells which do not exist in the constraints. In addition, the ML and CL constraints in the visualization in Figure 3 should be independent with the constraints used to train the scDCC model.

Response

We thank the reviewer for addressing this issue. In the revised manuscript, we applied a new method to generate constraints and evaluate the performance of different methods. For each dataset, we randomly selected 10% cells with labels to generate constraints. The scDCC model was trained on the dataset with constructed constraints. After obtaining clustering results, we evaluated the clustering metrics (NMI, CA and ARI) on the remaining 90% of cells. Therefore, in the new evaluation paradigm, there is no overlap between the cells for generating the pairwise constraints and the cells for evaluation. The cells used for building constraints didn't disclose any label information to the scDCC model (Results section "Pairwise constraints" and Methods section "Constraint Construction").

3. Analogously, triplet constraints were generated based on the embedding trained by scDCC with 20,000 pairwise constraints, which again indicates that the model obtained some true results in advance. It would be better if the authors pre-train scDCC without constraints to obtain the embedding.

Response

We thank the reviewer for raising this point. The triplet constraint is a weaker constraint than pairwise constraint. We have tried to select 10% cells to generate triplet constraints, but the improvement of performance is not very impressive comparing to the case without using triplet constraints. So, in the revised manuscript, we decide to remove the results of triplet constraint, and only focus on pairwise constraint.

4. It is good to see the authors evaluate the robustness of scDCC using 5% erroneous constraints. Given that the prior information may be very noisy in real application scenarios, I am interested in what percentage of erroneous constraints will invalidate this model?

Response

We thank the reviewer for raising this point. In the revised manuscript, we evaluated our methods on 5% and 10% erroneous constraints (Figure S9 and S11). As shown in Figure S9, with 10% erroneous constraints, scDCC on some datasets (worm neuron cells and human kidney cells) can still perform better with more constraints, but it began to perform worse on some datasets (mouse bladder cells). Therefore, the invalidating point may vary with different datasets (e.g., signal to noise ratio). Here we see that 10% of erroneous constraints may potentially invalidate the model and, therefore, users need to be cautious to move forward in such situations.

5. The model with combined constraints seems to use twice as many constraints as the one with only pairwise or triplet constraints. If so, the result of the comparison in Figure 6 may not be fair.

Response

We thank the reviewer for raising this point. The idea of triplet constraints was first inspired by computer vision research⁴. During the revision, we conducted a series experiments to systematically compare pairwise constraints and triplet constraints and find that the pairwise

constraints are usually better than triplet constraints in terms of the clustering performance. In addition, the pairwise constraints are more straightforward and easier to construct in different applications. Triplet constraint needs a predefined continuous prior information (e.g., a good latent representation of cells), which is not very feasible in the context of single cell analysis. Therefore, we focus our work on the pairwise constraints in the revised manuscript. Although we removed it from the main manuscript, we still provide the triplet constraint functionality in our software, which will be publicly available online.

6. In the example of marker gene-based constraints (Figure 7a), it would be better if the authors could compare the performance of Louvain clustering or k-means on the ZIFA latent representations of the marker gene matrix. Besides, it should be compared with the baseline methods.

Response

We thank the reviewer for raising this point. We added k-means clustering on the ZIFA latent representations of the marker genes and SC3 on top 5000 genes in the revised Figure 6a (Results section “Marker gene-based constraints”). As it shows, scDCC with constraints outperforms both PhenoGraph³ (Louvain clustering), k-means on marker genes and SC3. The measurements of scDCC and k-means are NMI: 0.905 vs 0.833, CA: 0.928 vs 0.928, ARI: 0.952 vs 0.924 (averages of ten repeats).

7. Considering that the authors listed a few cell type-specific signature sets as prior knowledges, it would be better if they can discuss in more detail about how to integrate these prior knowledges.

Response

We thank the reviewer for raising this point. In the revised manuscript, we added another case study (based on the “CITE-seq PBMC” dataset) to illustrate how to construct constraints and solve the clustering problem desirably. The key point is to generate accurate constraints and the definitions of the must-link and cannot-link faithfully represent the domain knowledge. We hope the two use cases listed in the manuscript could ignite the creativity of users to construct constraints for encoding their own domain knowledge.

Minor points:

1. <https://github.com/ttgump/scDCC> is unavailable.

Response

We have double checked the github to make sure that it could be freely downloaded.

2. In the last paragraph of Results, it should be ‘with 6000 triplet constraints (Figure 7e)’.

Response

The manuscript has been updated.

3. The authors overemphasized that scDCC does not require exact label information, since this is a clustering task where we seek to know which samples belong to the same cluster rather than a classification where we want to predict the exact label of each sample.

Response

We agree that the cluster label information can help scDCC to generate constraints. We show via two case studies in the revised manuscript (Results section “Protein marker-based constraints” and “Marker gene-based constraints”) that other data sources for labelling cell types/cell clusters, such as marker genes or cell surface proteins, can be incorporated into the model to improve clustering performance and provide desired and more interpretable results.

Reviewer #2 (Remarks to the Author):

This article introduces a deep neural network model for constrained clustering of single-cell RNAseq profiles. The constraints are based on log Bernoulli likelihood on connectivities between pairs or triples of samples. The methods are applied to several real single-cell RNAseq datasets. The paper is well-written and the methods are clearly described. There are some critical questions to address in the validation.

1. The clustering performance shown in Figure 2, 4, 5 and 6 are not properly evaluated since the training pairs are also included in the calculation of the measures. Similar to semi-supervised learning, the clustering of the constrained model can be very sensitive to any kind of supervised information. In principle, any samples that are presented with supervised information (such as pairwise relations in this model) should not be used in the evaluation. The proper experimental setup should be split the samples in the dataset into a training set and a test set. The constrained pairs/triples should only be sampled from the training set and the clustering results should only be reported for the samples in the test set. If there are a excessive number of hyper-parameters such as neural network structures, a validation set might also be needed. Given the model was only tested on a subset of each of the datasets, there should be enough samples for the train/test setup.

Without the correct validation, it is still unclear how much additional information is introduced by the pairs/triples to improve clustering in the current work.

Response

We thank the reviewer for raising this point and providing constructive suggestions. In the revised manuscript, we applied the method the reviewer suggested to evaluate the performance for all datasets. Specifically, for each dataset, we randomly selected 10% cells with labels to generate constraints. The scDCC model was trained on the dataset with constructed constraints. After obtaining clustering results, we evaluated the clustering metrics (NMI, CA and ARI) on the remaining 90% cells. Therefore, in the new evaluation paradigm, there is no overlap between the cells for generating the pairwise constraints and the cells for evaluation. In the revised figures, we apply the new evaluation method on all experiments including Figure 2,

Figure 4, Figure S1-8 and Figure S10 (Results section “Pairwise constraints” and Methods section “Constraint Construction”). Note that our method did not include hyper-parameter tuning, all results were obtained by the same set of parameters, which showed from another aspect that the proposed method is robust and useful.

2. The motivation of modeling the pairwise/tripwise relations in real studies is still unclear. The application to the five real datasets was taken by sampling from the final complete clustering in those studies, in which none of the dataset was originally generated with the pairwise knowledge. The common scenario is the case that a small set of cells are manually annotated and then used to train a model for classifying/clustering more single-cells from additional experiments, which is the case of supervised learning or semi-supervised learning. The pairwise constraint is still uncommon. It is important to justify the usefulness this model with real application scenarios, which is missing in the experiments or discussions.

Response

We thank the reviewer for raising this point. This is a semi-supervised clustering problem, which emphasizes clustering. It is different from semi-supervised classification, which is more in keeping with the semi-supervised learning the reviewer mentioned. In fact, pairwise constraint has been commonly considered in semi-supervised clustering^{4, 5, 6}. The motivation of modeling the pairwise/triplet relations has been illustrated in the computer vision researches⁴. We have similar motivation and justification for single cell studies. To illustrate the application of constraint in real scenarios, in the revised manuscript, we conducted two experiments: (a) CITE-seq PBMC data (Results section “Protein marker-based constraints”) and (b) marker genes of human liver cells (Results section “Marker gene-based constraints”).

- (a) CITE-seq can profile mRNA and protein expression levels simultaneously. To leverage the information of protein expression levels, we used a stringent method to generate constraints based on protein levels: we calculated Euclidean distances for all possible pairs of cells based on the normalized protein data and chose the 0.5th and 95th percentile of all pairwise distances as the must-link and cannot-link constraint cutoffs. 20,000 constraints were generated by this criterion. To further identify subtypes of CD4 and CD8 T cells (e.g., CD8+CD27-, CD8+CD27+, CD4+CD27+, CD4+CD27-DR+, CD4+CD27-DR-), we generate 5000 must-links for each subtype based on additional proteins. As shown in Figure 5, by incorporating constraints based on protein levels, scDCC can improve performance significantly, and scDCC with constraints can outperform PhenoGraph and k-means on protein levels.
- (b) In Figure 6, we conducted experiment of constraints generated by marker genes in human liver dataset. Marker genes can be considered as key features to separate different clusters. We first selected marker gene, then used ZIFA to reduce the dimension to 10. Next, we applied k-means on the ZIFA latent representations and used k-means results as the pseudo labels. ML and CL constraints were generated on these pseudo labels. As summarized in Figure 6, by incorporating constraints, scDCC not only improved performances (compared with scDCC without constraints), but also outperformed k-means on the ZIFA latent representations.

We think these two examples of constraints (generated on protein levels or marker genes) demonstrate the usefulness of this model with real application scenarios, yielding more interpretable results with improved clustering performance.

A minor note:

3. On paper 9, the definition of p_{ij} abuses the use of indexes j and j' in the summations, which creates some confusion in understanding the statistics (which might be an error from the original equation in citation 46).

Response

We thank the reviewer for raising this point. We have updated the equation.

References

1. Kiselev VY, *et al.* SC3: consensus clustering of single-cell RNA-seq data. *Nat Methods* **14**, 483-486 (2017).
2. Butler A, Hoffman P, Smibert P, Papalexi E, Satija R. Integrating single-cell transcriptomic data across different conditions, technologies, and species. *Nat Biotechnol* **36**, 411-420 (2018).
3. Levine JH, *et al.* Data-Driven Phenotypic Dissection of AML Reveals Progenitor-like Cells that Correlate with Prognosis. *Cell* **162**, 184-197 (2015).
4. Zhang H, Basu S, Davidson I. A Framework for Deep Constrained Clustering - Algorithms and Advances. *arXiv:190110061*, (2019).
5. Basu S, Davidson I, Wagstaff K. *Constrained Clustering: Advances in Algorithms, Theory, and Applications*. Chapman and Hall/CRC (2008).
6. Bilenko M, Basu S, Mooney RJ. Integrating constraints and metric learning in semi-supervised clustering. In: *Proceedings of the twenty-first international conference on Machine learning*. ACM (2004).

Reviewers' Comments:

Reviewer #1:

Remarks to the Author:

Thanks for addressing my concerns!

Reviewer #2:

Remarks to the Author:

The authors corrected the experimental design with proper training and test division of the data. The improvement is much appreciated.

In this revision, it is also helpful to add the CITE-seq PBMC data to justify the use of pairwise constraints derived from the single-cell protein markers. However, there are a few questions remain and more clarifications and possibly a better experimental design are needed in the analysis of the CITE-seq PBMC data.

1. How was the CITE-seq PBMC data labeled into 12 clusters? It appears the 12 clusters were derived from scRNAseq data analysis in the original study, which might not be used as true labels for the evaluation?
2. It appears the information from the 49 protein markers can already define the clusters. What happens with clustering with the 49 protein marker data only? What happens if use the markers to define the classes for supervised learning instead (this also related to the first point)? Without the comparisons, it is still unclear if the method is useful or not in this scenario when both scRNAseq and surface protein markers are measured with CITE-seq.

Reviewer #2 (Remarks to the Author):

The authors corrected the experimental design with proper training and test division of the data. The improvement is much appreciated.

In this revision, it is also helpful to add the CITE-seq PBMC data to justify the use of pairwise constraints derived from the single-cell protein markers. However, there are a few questions remain and more clarifications and possibly a better experimental design are needed in the analysis of the CITE-seq PBMC data.

1. How was the CITE-seq PBMC data labeled into 12 clusters? It appears the 12 clusters were derived from scRNAseq data analysis in the original study, which might not be used as true labels for the evaluation?

Response

For all the datasets used in the paper, including the CITE-seq PBMC dataset, we used the cluster labels developed for the original study. The labeling procedure, which is the same or similar across different studies, consists of two major steps. First, initial clustering: perform clustering analysis using a popular scRNA-seq tool Seurat on the mRNA counts. Second, perform manual adjustment and labelling as follows: (a) conduct differential expression (DE) analysis for each cluster (one cluster vs the others); (b) then check the canonical marker genes of known cell types to see if they are significantly highly expressed in the cluster; if so, then label it to the known cell type; (c) the clusters having the same label will be merged into one cluster. We note some minor differences between different studies, such as the use of Seurat vs. k-means as the criteria for claiming the set of marker genes of being significantly highly expressed, etc.

Such a labelling procedure is being widely used in the field, which is perhaps the best ground truth labelling we could have¹. Most studies developing computational methods use the labels generated in the original study and work on the same data. The evaluation rationale is whether a proposed method can generate optimal clustering results in a principled way without going through multiple heuristic processing steps. The clustering performance metrics based on such labelling would help to give us some quantitative measurement of the performance of clustering algorithms. As such, we are following the convention in the field for the labelling approach.

To further demonstrate the contribution of constraints, we visualized the CD4 and CD8 protein levels in the clustering results (**Figure 5b**). We can see that, with the help of constraint information, marker genes CD4 and CD8 are separated well into different clusters, which makes good biological sense and is desired by biologists. We use this as a complementary evidence to support the superiority of the proposed method.

2. It appears the information from the 49 protein markers can already define the clusters. What happens with clustering with the 49 protein marker data only?

Response

(a) We have done PhenoGraph (Louvian clustering) and k-means using the 49 protein markers to define the clusters. Indeed, their performances are better than clustering results of SC3 and scDeepCluster on mRNAs (**Figure 5a**), which indicates that the information from the 49 protein markers can already define the clusters to a good extent. However, there is room for improvement. As we can see, scDCC with constraints can improve over these baseline methods, which illustrates the contribution of the proposed method.

What happens if use the markers to define the classes for supervised learning instead (this also related to the first point)? Without the comparisons, it is still unclear if the method is useful or not in this scenario when both scRNAseq and surface protein markers are measured with CITE-seq.

Response

(b) It is unclear what kind of ‘supervised’ learning the reviewer is suggesting because the task here is clustering, which does not provide labelled samples for training. However, we considered the following approach to define classes and conducted some supervised learning experiments, which we hope addresses the reviewer’s concern.

To obtain initial labels, we performed k-means clustering based on the 49 protein markers to define clusters as in (a) (Note, PhenoGraph is not used because it does not provide centroids which we need to use later). Such initial labeling (clustering) may not be optimal, however, so we keep the labels for only the top x% cells closest to their cluster centroids assuming that for a sample, the closer to the centroid, the more accurate is its assigned label. Then we use these x% ‘labelled’ cells as training samples to train a predictive model to predict the remaining (1-x%) unlabeled cells using the mRNA data. Then we have clustering labels for all samples. When x% is too small we may not have enough training samples to train the predictive model well. When x% is large, fewer samples will benefit from the supervised learning model. In an extreme case, when x% =100%, it is essentially what we do in (a). Given this tradeoff, we set x%=50%. We choose SVM as the predictive model.

Methods	NMI	CA	ARI
scDCC with constraints	0.714	0.747	0.701
Supervised learning (SVM)	0.663	0.708	0.611

As we can see from the Table above, scDCC with constraints still outperforms the supervised learning approach. This observation is consistent with existing literatures that supervised learning methods may be ineffective when labeled data is scarce^{2, 3, 4, 5}. It is noted that class labels (used by supervised learning) can also be translated into pairwise constraints for the labeled cells; vice versa, given consistent pairwise constraints for some cells, we may derive group information for the cells with constraints. Nevertheless, our proposed method is more flexible and effective, especially when the available knowledge (i.e., cell label) is too far from

being representative of a target classification of the cells and supervised learning is not effective or possible, even in a transductive form².

References

1. Kiselev VY, Andrews TS, Hemberg M. Challenges in unsupervised clustering of single-cell RNA-seq data. *Nat Rev Genet* **20**, 273-282 (2019).
2. Grira N, Crucianu M, Boujemaa N. Unsupervised and Semi-supervised Clustering: a Brief Survey. *A Review of Machine Learning Techniques for Processing Multimedia Content* **1**, 9-16 (2004).
3. Bair E. Semi-supervised clustering methods. *Wiley Interdiscip Rev Comput Stat* **5**, 349-361 (2013).
4. van Engelen JE, Hoos HH. A survey on semi-supervised learning. *Machine Learning* **109**, 373-440 (2019).
5. Joachims T. Transductive Inference for Text Classification using Support Vector Machines. In: *Proceedings of the Sixteenth International Conference on Machine Learning (ICML 99)* (1999).

Reviewers' Comments:

Reviewer #1:

None